# A Strip-Till One-Pass System as a Component of Conservation Agriculture

**Iwona Jaskulska [1],\*, Kestutis Romaneckas [2], Dariusz Jaskulski [1] and Piotr Wojewódzki [3]**

[1] Department of Agronomy, Faculty of Agriculture and Biotechnology,
UTP University of Science and Technology, 7 prof. S. Kaliskiego St., 85-796 Bydgoszcz, Poland;
darekjas@utp.edu.pl

[2] Institute of Agroecosystems and Soil Sciences, Vytautas Magnus University, Agriculture Academy,
K. Donelaičio str. 58, 44248 Kaunas, Lithuania; kestutis.romaneckas@vdu.lt

[3] Department of Biogeochemistry and Soil Science, UTP University of Science and Technology,
Bernardyńska 6/8 Street, 85-029 Bydgoszcz, Poland; pwoj@utp.edu.pl

\* Correspondence: jaskulska@utp.edu.pl

**Abstract:** Conservation agriculture has three main pillars, i.e., minimum tillage, permanent soil cover, and crop rotation. Covering the soil surface with plant residues and minimum mechanical soil disturbance can all result from introducing a strip-till one-pass (ST-OP) system. The aim of this study was to determine the impact of the ST-OP technology on the management of plant residues, soil properties, inputs, and emissions related to crop cultivation. We compared the effect of a ST-OP system against conventional tillage (CT) using a plough, and against reduced, non-ploughing tillage (RT). Four field experiments were conducted for evaluating the covering of soil with plant residues of the previous crop, soil loss on a slope exposed to surface soil runoff, soil structure and aggregate stability, occurrence of soil organisms and glomalin content, soil moisture and soil water reserve during plant sowing, labour and fuel inputs, and $CO_2$ emissions. After sowing plants using ST-OP, 62.7–82.0% of plant residues remained on the soil surface, depending on the previous crop and row spacing. As compared with CT, the ST-OP system increased the stability of soil aggregates of 0.25–2.0 mm diameter by 12.7%, glomalin content by 0.08 g·kg$^{-1}$, weight of earthworms five-fold, bacteria and fungi counts, and moisture content in the soil; meanwhile, it decreased soil loss by 2.57–6.36 t·ha$^{-1}$ year$^{-1}$, labour input by 114–152 min·ha$^{-1}$, fuel consumption by 35.9–45.8 l·ha$^{-1}$, and $CO_2$ emissions by 98.7–125.9 kg·ha$^{-1}$. Significant favourable changes, as compared with reduced tillage (RT), were also found with respect to the stability index of aggregates of 2.0–10.0 mm diameter, the number and weight of earthworms, as well as bacteria and fungi counts.

**Keywords:** conservation agriculture; strip-till; plant residues; soil erosion; soil structure; soil organisms; glomalin; fuel consumption; $CO_2$ emission

## 1. Introduction

According to the FAO definition [1], conservation agriculture is a system that promotes minimal tillage and the covering of soil with vegetation and/or mulch. Reducing tillage is one of the main agrotechnical methods for increasing the retention of organic carbon in the soil [2], improving physical properties [3,4], increasing the water penetration resistance of soil aggregates and the stability of soil structure [5,6], and reducing erosion and nutrient depletion [7,8]. Soil tillage systems can be classified in descending order in terms of their propensity to facilitate soil erosion as follows: conventional tillage > reduced tillage > no tillage [9].

The strip-till system fulfils the principles of conservation agriculture. Only narrow strips of soil are loosened, though to a considerable depth. They constitute a maximum of one third of the field area.

A minimum of 50–75% of the soil is covered with plant residues [10–12]. Using a strip-till one-pass technique, fertilisers and seeds are applied during the loosening of soil strips. Additional agrotechnical practices can be performed at the same time, for example, applying pesticides amd sowing intercrops. This is more economical and reduces the pressure that field crop production exerts on the environment [13].

Zonal tilling, including strip-till, is appropriate for sustainable farming systems. Specialised machines are now allowing strip-till one-pass to be used to cultivate crops with wide [14,15], medium [16,17], and narrow row spacing [18,19]. It combines the advantage of conventional tillage, plough tillage, and no-till systems. In the horizontal plane, it creates a pattern of narrow strips of loosened, but not inverted soil, and strips of unloosened soil. Hydrothermal conditions and the nutrient availability differ between tilled rows and non-tilled inter-rows. After zonal tillage, moisture is higher in the unloosened soil of the inter-rows than in tilled soil [20,21]. Within the rows, the soil is deeply loosened, and has low bulk density and penetration resistance, but a high-water infiltration rate. In the inter-rows, the soil is covered with plant residues; it is moist and has a lower temperature [22]. The plant residues and mulch reduce surface water runoff, protect the structure. and, by lowering soil temperature, reduce evaporation [23,24]. Meanwhile, in the vertical plane, soil properties are differentiated into layers, especially after turning the soil with a plough is replaced with loosening and no-tillage. Organic matter and nutrients accumulate in the surface layer [25], affecting soil structure and fertility [26]. The relative increase in organic carbon in the surface layer can be up to 10% within a few years [27]. The horizontal and vertical changes in the physical and chemical properties of the soil are also associated with changes in biological and biochemical properties [28,29], including the presence of earthworms and glomalin content. Soil tillage can alter earthworm abundance in the soil by a factor of between two and nine [30,31]. Glomalin is a glycoprotein secreted from hyphae of arbuscular mycorrhizal fungi (AMF) in the phylum Glomeromycota [32]. AMF occur on the roots of 80% of vascular plants [33]. Many crops are dependent on AMF mycorrhiza, for example, corn, legumes, bean and potatoes, while others like wheat, oat, or barley just benefit from it without strong dependence [34]. Glomalin concentration in soils vary widely depending on land uses [35]. Observations indicate that its content decreases due to frequent agrotechnical treatments [36,37] or due to increasing soil salinity [38].

Conservation agriculture, if selected appropriately to local environmental conditions, positively affects the complex of soil properties, including productivity and crop yields [39]. Reduced soil tillage or direct sowing allow inputs to be reduced and similar or even higher arable crop yields to be achieved as compared with conventional tillage. According to Gozubuyuk et al. [40], after replacing conventional tillage with reduced tillage, fuel consumption was 3.5 times lower, and $CO_2$ emissions fell logarithmically in proportion to the reduction in soil tillage. Reduced tillage may, however, result in higher $CO_2$ emissions when plant residues are heavily mineralised on the soil surface [41]. Thus, more research is needed on the environmental impact of agriculture, especially in view of the introduction and diffusion of new technologies.

The scientific literature lacks research results, especially from studies performed in Central European habitat and agronomic conditions and relating to the effect that strip-till one-pass has on soil properties, fuel consumption, and $CO_2$ emissions. There is a need to verify the hypothesis that the retention of plant residues on the soil surface and changes in soil properties that this method encourages cause a reduction in erosion, an increase in structural stability, greater abundance of soil organisms, improved hydrological conditions, and reduced inputs and $CO_2$ emissions, in agricultural crop production. The confirmation of this hypothesis would allow strip-till one-pass to be propagated as a component of conservation agriculture.

The field-experiment-based studies aimed to determine the influence of strip-till one-pass on the management of plant residues, soil physical and biological properties, labour, and fuel inputs on crop cultivation and on fuel-related $CO_2$ emissions.

## 2. Materials and Methods

### 2.1. Study Site

The research was carried out on a farm more than 1000 ha and the *Agro-Środki-Technika-Technologia* Research and Development Centre owned by Marek Różniak, Agro-Land Śmielin (53°09′04.0″ N, 17°29′10.7″ E, 93.8 m a.s.l.), Kuyavian-Pomeranian Voivodeship, Poland. The farm and the Research and Development Centre cooperate with the Department of Agronomy at the University of Technology and Life Sciences in Bydgoszcz (UTP University, Bydgoszcz, Poland).

The farm has over 1000 ha of soils classified as *Luvisols* and *Cambisols* [42]. According to the Köppen classification, the research area lies in a humid continental climate zone [43]. The site meteorological conditions (precipitation and air temperature) for the entire study period were presented in an earlier paper [44]. The average annual sum of precipitation is 485 mm. Most precipitation occurs in July and August, with more than 60 mm. The average monthly mean air temperature is 8.1 °C; the hottest month is July and the coldest is January. Table 1 shows the weather conditions during the experimental period (2015–2019).

**Table 1.** Meteorological conditions in 2015–2019 at the site of field experiments.

| Month | Year | | | | |
|---|---|---|---|---|---|
| | 2015 | 2016 | 2017 | 2018 | 2019 |
| Air temperature (°C) | | | | | |
| January | 1.1 | −3.3 | −2.6 | 0.8 | −0.7 |
| February | 0.1 | 2.5 | −0.5 | −3.2 | 2.6 |
| March | 4.1 | 33.0 | 5.4 | −0.2 | 5.4 |
| April | 7.6 | 8.3 | 6.8 | 12.0 | 9.3 |
| May | 12.4 | 14.7 | 13.4 | 16.9 | 12.1 |
| June | 15.6 | 17.7 | 16.8 | 18.4 | 21.9 |
| July | 18.5 | 18.3 | 17.7 | 20.5 | 18.6 |
| August | 21.0 | 16.4 | 17.7 | 19.9 | 19.7 |
| September | 13.8 | 14.3 | 13.0 | 15.6 | 13.5 |
| October | 6.4 | 6.3 | 10.1 | 9.8 | 9.8 |
| November | 4.8 | 2.5 | 4.5 | 4.5 | 5.5 |
| December | 3.7 | 1.4 | 2.0 | 2.0 | 2.7 |
| Average | 9.1 | 11.0 | 8.7 | 9.8 | 10.0 |
| Precipitation (mm) | | | | | |
| January | 33.2 | 20.3 | 14.5 | 46.3 | 32.6 |
| February | 8.9 | 19.0 | 30.3 | 5.8 | 18.1 |
| March | 35.7 | 23.2 | 27.5 | 16.4 | 28.8 |
| April | 15.6 | 28.7 | 40.8 | 40.4 | 1.5 |
| May | 21.6 | 51.4 | 56.3 | 14.2 | 89.2 |
| June | 33.0 | 98.1 | 54.3 | 26.4 | 17.7 |
| July | 50.4 | 133.8 | 118.9 | 86.0 | 22.4 |
| August | 20.3 | 55.3 | 126.1 | 23.7 | 37.7 |
| September | 52.4 | 19.4 | 78.4 | 17.0 | 98.5 |
| October | 20.9 | 116.3 | 106.8 | 34.1 | 35.9 |
| November | 37.0 | 41.7 | 30.5 | 7.2 | 69.6 |
| December | 24.4 | 42.7 | 38.8 | 50.3 | 21.1 |
| Sum | 353.4 | 649.9 | 723.2 | 367.8 | 473.1 |

### 2.2. Design and Performance of Field Experiments

The impact of the strip-till one-pass (ST-OP) system on soil properties and the implementation of conservation agriculture principles were assessed by field study. To this end, scientific field experiments were carried out in production farm conditions, and a ST-OP system was compared against conventional

plough tillage (CT), and against conventional (CT) and reduced, non-ploughing tillage (RT) in terms of their effects on the following:

i.　　Management of crop residue after previous crop harvesting;
ii.　　Surface soil runoff, soil loss on a slope;
iii.　　Soil structure and glomalin content;
iv.　　Number and mass of earthworms, total bacteria and fungi count;
v.　　Spatial differentiation of soil moisture and soil water reserve during sowing and seedling emergence;
vi.　　Labour input, fuel consumption, and $CO_2$ emissions.

2.2.1. Experiment 1

In each of the three crop-growing seasons (2016/17–2018/19) seven crop species were sown on the experimental demonstration fields. Each species (winter wheat, winter barley, spring barley, soybean, winter rape, sunflower, and maize) was sown in triplicate on plots of 300 m long and 12 m wide. After harvesting, the grain or seeds and shredding the straw of each of the seven crops on each of the plots, which were divided into three 4 m wide belts (sub-plots), the following were cultivated: wheat, with a 15 cm row spacing after mouldboard ploughing, wheat by ST-OP method, and maize by the same method (Figure 1). This, after harvesting each pre-crop, provided the following three experimental treatments for the management of plant residues:

1.　　Conventional technology (CT) with mouldboard ploughing turning over plant residues (with wheat as the subsequent crop);
2.　　Strip-till one-pass method ST-OP 36 (with wheat as the subsequent crop),
3.　　Strip-till one-pass method ST-OP 72 (with maize as the subsequent crop).

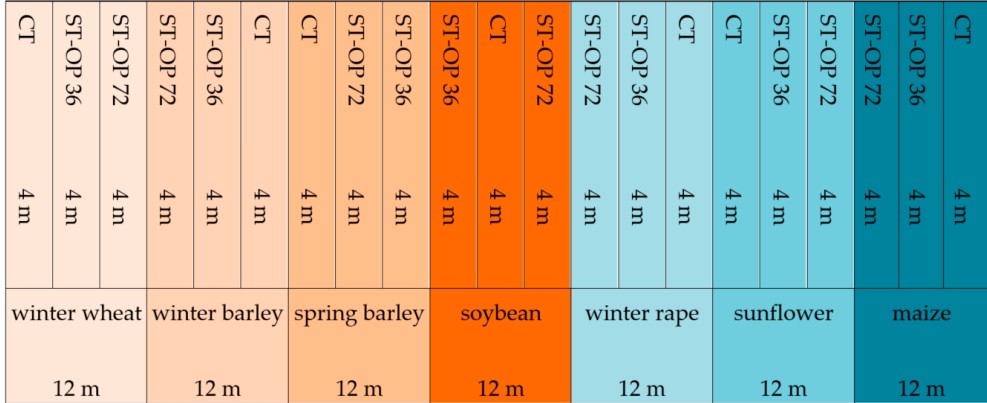

**Figure 1.** Scheme of arrangement of previous crops and succeeding one to estimate the impact of the tillage method on plant residues, one of three replications. CT, conventional; ST-OP 36, strip-till one-pass, 36 cm strip spacing; ST-OP 72, strip-till one-pass, 72 cm strip spacing.

The ST-OP cultivation was performed using a Mzuri Pro-Til hybrid machine. The machine cooperated with the tractor Claas Axion 850, speed about 7–8 km/h. Wheat was sown in two rows in 12 cm strips of loosened soil with 36 cm row spacing. The machine's loosening tines and sowing coulters loosen the soil and mix it with the plant residues to form small contour ridges in the unloosened 24 cm wide inter-row (Figure 2A). Maize was sown in a single row in the middle of the loosened row at a 72 cm row spacing. In the middle of the 60 cm inter-row, next to the "microridges" a layer of mulch was left consisting of residues of the previous crop (Figure 2B).

After harvesting the previous crop and marking out the experimental treatments in five locations (each of 1 m$^2$) in each plot, the dry weight of plant residue was collected and determined by air drying. The same method was used to determine the weight of plant residue immediately after crop sowing.

The difference between weight of plant residues after harvesting and that after sowing was used to determine the percentage of residues remaining as mulch on the soil surface. In this way, the impact of soil cultivation technology, pre-sowing fertilization and sowing, on plant residues of the pre-crop and soil mulching was assessed.

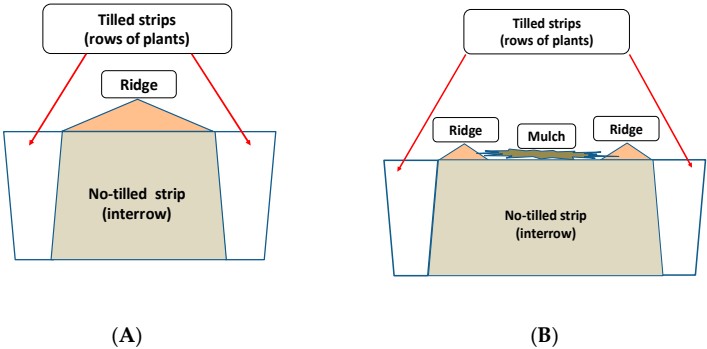

(**A**)      (**B**)

**Figure 2.** Field area after sowing plants in the strip-till one-pass technology (scheme) using the Mzuri Pro-Til hybrid machine. (**A**) With a tilled strip spacing of 36 cm; (**B**) With a tilled strip spacing of 72 cm

### 2.2.2. Experiment 2

On an 11% slope (Figure 3A), the effects of CT (conventional plough tillage and conventional row drilling) after seedbed preparation were compared against the effects of ST-OP plant cultivation as two experimental treatments.

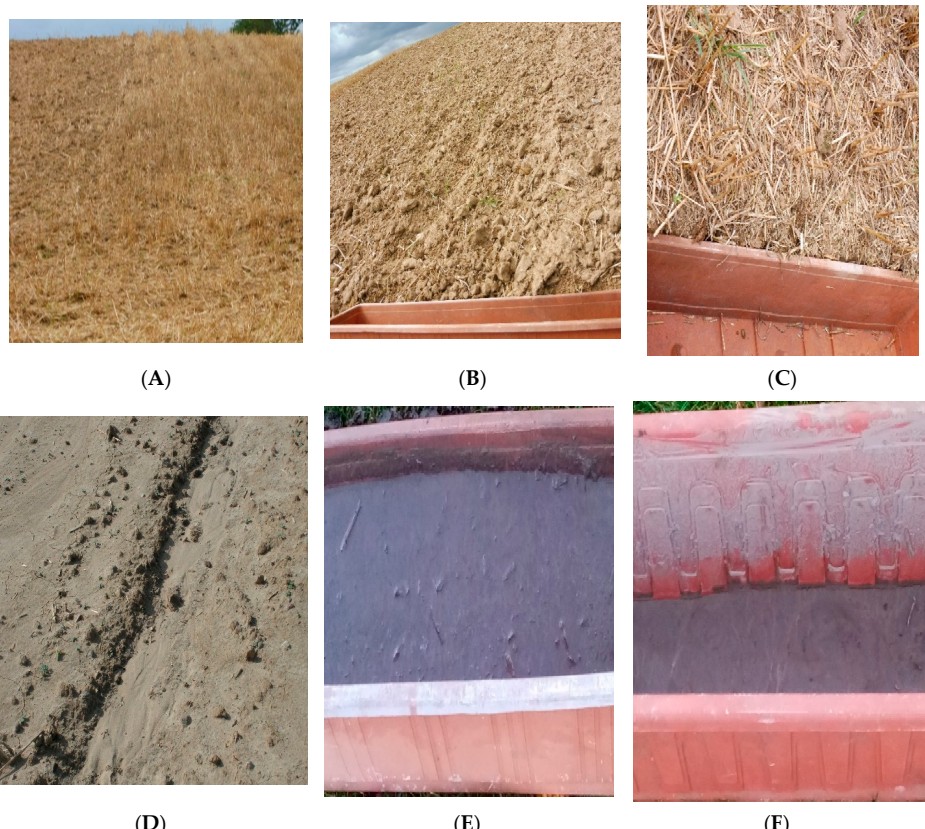

**Figure 3.** The site of the surface runoff and soil loss experiment. (**A**) Field on a slope of 11% gradient; (**B**) Sedimentation container on a ploughed slope; (**C**) Sedimentation container on a mulched slope (strip-till one-pass, ST-OP); (**D**) Field surface after storm (erosion), conventional cultivation (CT); (**E**) Soil lost from CT field; (**F**) Soil lost from ST-OP.

The experiment was performed in the years 2015–2018 on sandy loam soil of grain size as follows: sand (2–0.05 mm) 48.2%, silt (0.05–0.002 mm) 47.0%, clay (<0.002 mm) 4.8%. In successive years of the experiment, the crops grown were winter rape, winter wheat, and maize. For both experimental treatments, plots were designated to quantify surface runoff and soil loss. Plots of 15 m long and 1.5 m wide were designated running lengthwise down the slope, and a sedimentation container for eroded soil was installed at foot of the slope (Figure 3B,C). This allowed for quantification of the amount of soil loss (soil dry matter) from the conventionally tilled area and the ST-OP areas both after heavy rainfalls (Figure 3D) and over the cultivation period (Figure 3E,F).

### 2.2.3. Experiment 3

In 2019, after crops were harvested, soil samples were taken from the three sites of the static long-term field experiment. This experiment was set up in 2012 and described in detail in a previous paper [44]. Soil samples of 3 kg were collected from five locations randomly selected along the diagonals of each plot. After seven years of conventional tillage (CT), non-ploughed reduced tillage (RT) and strip-till one-pass (ST-OP), as a long-term experiment factor, we determined the number of total bacteria and filamentous fungi, the number and mass of earthworms, and the soil structure in the 0–20 cm layer (aggregate size distribution (ASD), mean weight diameter of soil aggregates (MWD), soil aggregate stability (SAS)), and glomalin content (easily extractable glomalin-related soil protein (EEGRSP)).

The number and mass of earthworms were determined in the field in $15 \times 15 \times 20$ cm soil monoliths. The earthworms were placed back into the soil immediately after evaluation. The microorganisms were evaluated in the laboratory after adding Ringer's solution and shaking soil samples for 30 min. Then, series of decimal dilutions ($10^{-1}$ to $10^{-7}$) were performed. From the soil solutions, the following inoculations were made into a culture media:

- YPS with soil extract (incubation for 5 days at 26 °C) added to evaluate total bacteria count;
- The Martin medium with 30 µg mL$^{-1}$ streptomycin added to evaluate the filamentous fungi (incubation for 5 days at 25 °C).

The number of colony-forming units (CFU) was converted to number per 1 g of soil (CFU·g$^{-1}$ of soil).

The remaining soil samples were dried in the laboratory at 20–22 °C, then, dry separated (Retsch vibratory sieve shaker AS 200 with a set of sieves with mesh diameters 0.25, 0.50, 1.0 2.0, 3.0, 5.0, 7.0 and 10.0 mm). On the basis of these results, the aggregate size distribution (ASD) and mean weight diameter of aggregate (MWD) were determined according to the Equation (1) as:

$$\text{MWD} = \sum \text{xi wi [mm]} \tag{1}$$

where MWD is the mean weight diameter, xi is the mean diameter of the i-th sieve size, and wi is the proportion of the total aggregates in the i-th fraction.

Stability of soil aggregates (0.25–2.0 mm and 2.0–10.0 mm) was measured in a wet-sieving apparatus (Eijkelkamp 08.13) on 0.25 mm screens, according to the manufacturer's operating instructions. Stability of soil aggregates was tested in three replicates of ca 3–4 g samples. The air-dried aggregates were pre-moistened before submerging. Then, the sieves were placed in the sieve holder of the apparatus over vessels containing distilled water. Wet sieving was realised through 3 min (stroke 1.3 cm and oscillations 34 per minute). Soil particles that passed through the sieve were dried at 105 °C and weighed. The resistant soil material on each sieve was dispersed by solution of NaOH (2 g·dm$^{-3}$), dried at 105 °C, and weighed. The stable fraction is equal to the weight of material obtained in the dispersing solution vessels (A) divided by the sum of weights obtained in the dispersing solution and

distilled water vessels (B). The mass of NaOH was subtracted from the weight of dried, dispersed soil material. The soil aggregate stability index (SAS) was calculated on the base of Equation (2) as:

$$SAS = \frac{A}{A + B} \cdot 100 \; [\%] \tag{2}$$

Easily extractable glomalin-related soil protein (EEGRSP) was extracted, according to the method established by Wright and Upadhyaya [45]. The extraction protocol included the following: weighing ca 1 g of soil and placing it in a 50 mL PP centrifuge tube, adding 20 mM sodium citrate (pH 7.0), autoclaving (30 min., t = 121 °C, $p$ = 1.4 kg·cm$^{-1}$), centrifuging (15 min, 5000× $g$), and decanting of the supernatant, which was stored at 4 °C for further analysis. The EEGRSP content in supernatant was assayed by Bradford method with ready solutions (Bio-Rad 500-0207) of bovine serum albumin as the standard. Extracts were pipetted into disposable half-micro (50 µL) cuvettes and diluted by phosphate buffer saline (PBS) pH 7.4 (AppliChem A91770100). Then, the Bio-Rad protein dye reagent (Bio-Rad 500-0006) was added. After 5 min of incubation, the cuvettes were placed in the spectrophotometer.

### 2.2.4. Experiment 4

In the years 2016–2019, a fourth field experiment was conducted independent of Experiments 1–3. In three growing seasons for winter and spring crops, the impact that conventional, plough tillage has on soil water reserve during the winter barley and maize sowing periods was compared against the impact of ST-OP method. Each of the two experimental tillage treatments was performed in three repetitions on 100 m long by 12 m wide plots. Soil moisture in the cultivated layer (0–20 cm) was assessed for a month during the soil preparation and sowing of spring plants in the study area, thus, from April 20 to May 20 for maize, and from August 20 to September 20 for winter crops (barley). Soil moisture was measured once a week by time domain reflectometry method using a FieldScout TDR. Measurements were made randomly at 20 sites in each plot. The spatial differentiation of soil moisture in rows and in inter-rows was also determined immediately after sowing plants by strip-till method (5–7 days after sowing). Soil moisture was measured within the following: rows, the formed contour ridges, the inter-rows (beneath contour ridges), and the inter-rows (beneath the mulch). The soil water reserve in the tilled soil layer (W), averaged over the soil preparation and sowing period, was calculated by Equation (3) as:

$$W = \frac{F \, h \, m \, \rho w}{100000} \left( \frac{t}{ha} \right) \tag{3}$$

where F is area, 10,000 m$^2$; h isthickness of soil layer, 0.2 m; m issoil volume moisture, %; and $\rho w$ is water bulk density, 1000 kg·m$^{-3}$. The result expressed in t·ha$^{-1}$ was also converted into rainfall equivalent in mm.

In Experiment 4, the labour input and fuel consumption for agrotechnical activities related to soil tillage, pre-sowing fertilization, and sowing were assessed for winter barley and maize. The real duration of individual agricultural activities using agricultural machinery and tools was measured. Fuel consumption on large-area experimental plots was determined according to the records on a tractor's on-board computer. On the basis of fuel consumption, the amounts of emitted $CO_2$ were calculated. According to the American Petroleum Institute [46], consuming 1 litre of diesel results in the emission of 2.75 kg of $CO_2$.

### 2.3. Data Analysis

The research results were analysed mathematically and statistically. The values of measurable characteristics in the factorial field experiments were assessed for normality of distribution assuming the null hypothesis that the variables were normally distributed. This evaluation was performed using the Shapiro–Wilk test. Normally distributed data was subjected to ANOVA. The statistical

significance of the influence of the experimental treatments was assessed with the F test, and the significance of differences between mean values of individual features with the post-hoc Tukey's test at $p < 0.05$. In Experiment 1, in addition to assessing the significance of differences between absolute values, the relative values (%) of the mass of plant residues under the influence of various tilling and sowing methods were also determined. The weight of residues immediately after harvest was adopted as the 100% value. In Experiment 4, different zones of the tilled soil layer were used as experimental treatments for statistical evaluation of soil moisture directly after sowing winter barley and maize by the strip-till one-pass method. The results are shown as a diagram of the spatial differentiation of soil moisture after using the Mzuri Pro-Til machine with the ST-OP method. Statistical analyses were performed using Statistica 12 software [47].

## 3. Results

### 3.1. Experiment 1

The greatest amount of plant residues on the soil surface, in excess of 10 t·ha⁻¹, was left after the harvest of maize grain and sunflower seeds (Table 2). Soil inversion with a plough in conventional tillage best covered the plant residue of each pre-crop. On the field surface there remained from 3.2% of plant residues for spring barley and soybean, to 5.3% and 5.4% for corn and sunflower, respectively. The compared methods of tillage and crop sowing can be sorted in increasing order by mass of residues remaining on the field surface, as follows: CT < ST-OP 36 < ST-OP 72. The exception is the ST-OP method with different spacing of seed strips (36 cm and 72 cm) applied after the harvesting of spring barley and sunflower. After the two pre-crops, the mass of plant residues did not differ significantly between winter wheat (36 cm spacing of seed strips (ST-OP 36)) and maize (72 cm spacing (ST-OP 72)).

**Table 2.** Plant residues mass on the soil surface after the harvest previous crop (PC) and after various methods of soil tillage and the sowing. (conventional, CT, strip-till one-pass 36 cm, ST-OP 36; strip-till one-pass 72 cm, ST-OP 72).

| Crop | Previous Crop | | Conventional—CT | | ST-OP 36 | | ST-OP 72 | |
|---|---|---|---|---|---|---|---|---|
| | t ha⁻¹ | % | t ha⁻¹ | % | t ha⁻¹ | % | t ha⁻¹ | % |
| Winter wheat | 9.07 a | 100 | 0.37 d | 4.1 | 5.79 c | 63.8 | 6.89 b | 76.0 |
| Winter barley | 8.16 a | 100 | 0.30 d | 3.7 | 5.34 c | 65.4 | 6.30 b | 77.2 |
| Spring barley | 5.61 a | 100 | 0.18 c | 3.2 | 3.88 b | 69.2 | 4.25 b | 75.7 |
| Soybean | 5.31 a | 100 | 0.17 d | 3.2 | 3.35 c | 63.0 | 4.15 b | 78.2 |
| Winter rape | 8.94 a | 100 | 0.37 d | 4.1 | 5.61 c | 62.7 | 6.83 b | 76.4 |
| Sunflower | 10.23 a | 100 | 0.55 c | 5.4 | 7.64 b | 74.7 | 8.15 b | 79.7 |
| Maize | 15.65 a | 100 | 0.83 d | 5.3 | 10.96 c | 70.0 | 12.83 b | 82.0 |

The letters a–d indicate significant difference at $p < 0.05$.

### 3.2. Experiment 2

Surface water runoff and soil loss on the sloping field differed in intensity in successive years. In each of the three years of the study, the soil loss from the field surface cultivated using this technology was significantly lower than with conventional plough tillage (Figure 4). The quantitative difference in soil loss from these fields in the growing seasons of winter rape (2015/16), winter wheat (2016/17), and maize (2018) was 4.71 t·ha⁻¹, 6.36 t·ha⁻¹, and 2.57 t·ha⁻¹, respectively.

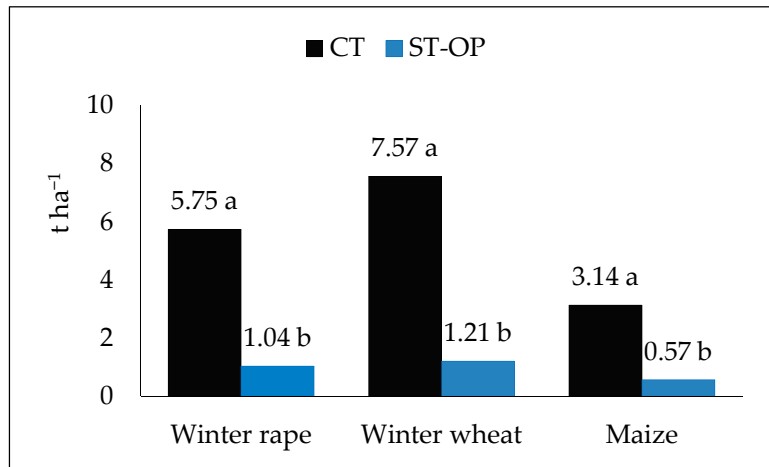

**Figure 4.** Soil loss during the growing season of winter rape (2015/2016), winter wheat (2016/2017), and maize (2018) depending on the method of soil tillage. Conventional (CT) and strip-till one-pass (ST-OP). The letters a and b indicate significant difference for tillage methods at $p < 0.05$.

*3.3. Experiment 3*

Long-term cultivation of soil and plants using the ST-OP system resulted in favourable changes in soil structure (Table 3). The share of aggregates of individual size classes (from 0–0.25 mm to >10.0 mm) was more evenly distributed (i.e., there was a lower coefficient of variation (CV)), and the MWD of the aggregate was significantly greater than was the case for the conventionally tilled (ploughed) soil. The soil structure aggregates also exhibited greater water penetration resistance. The soil aggregate stability index (SAS) for small aggregates was significantly higher than that of conventionally tilled soil aggregates. Large aggregates (2.0–10.0 mm) of the soil tilled by ST-OP technology were more stable than those cultivated conventionally or according to the RT system.

**Table 3.** The aggregate size distribution (ASD), mean weight diameter (MWD), and soil aggregate stability index (SAS) values, as a result of different tillage systems. Conventional (CT), reduced (RT), and strip-till one-pass (ST-OP).

| Soil Aggregates | Unit | Conventional (CT) | Reduced (RT) | Strip-Till One-Pass (ST-OP) |
|---|---|---|---|---|
| Aggregate size distribution (ASD) | | | | |
| <0.25 mm | % | 11.3 | 9.5 | 10.2 |
| 0.25–0.50 mm | % | 11.1 | 11.2 | 10.8 |
| 0.50–1.0 mm | % | 13.4 | 10.7 | 11.0 |
| 1.0–2.0 mm | % | 12.0 | 12.7 | 11.8 |
| 2.0–3.0 mm | % | 14.1 | 15.1 | 14.0 |
| 3.0–5.0 mm | % | 16.6 | 17.2 | 16.5 |
| 5.0–7.0 mm | % | 13.7 | 14.2 | 13.4 |
| 7.0–10.0 mm | % | 7.8 | 9.4 | 12.3 |
| CV | % | 20.8 | 22.5 | 16.6 |
| Mean weight diameter (MWD) | mm | 2.84 b | 3.04 a | 3.17 a |
| Soil aggregate stability index (SAS) | | | | |
| 0.25–2.0 mm | % | 52.4 b | 63.5 a | 65.1 a |
| 2.0–10.0 mm | % | 41.3 c | 49.6 b | 53.2 a |

The letters a, b, and c indicate significant difference at $p < 0.05$.

After many years of using the ST-OP method, the soil contained significantly more glomalin (EEGRSP) than did the conventionally tilled soil (Table 4). Simplifying soil tillage (RT), and ST-OP

technology in particular, resulted in increased bacteria and fungi counts. The ST-OP method was particularly favourable to the presence of earthworms in the arable soil layer. It resulted in more than three times more earthworms, totalling five times greater mass, than did conventional soil tillage.

**Table 4.** Earthworms, microorganisms, and glomalin (EEGRSP) content in soil as a result of different tillage systems. Conventional (CT), reduced (RT), and strip-till one-pass (ST-OP).

| Property | Unit | Conventional (CT) | Reduced (RT) | Strip-Till One-Pass (ST-OP) |
| --- | --- | --- | --- | --- |
| Glomalin (EEGRSP) | g kg$^{-1}$ | 0.85 b | 0.89 ab | 0.93 a |
| Earthworms | no m$^{-2}$ | 23.4 c | 39.7 b | 75.1 a |
| Earthworms | g m$^{-2}$ | 19.8 c | 45.2 b | 96.7 a |
| Bacteria | $10^6$ cfu g$^{-1}$ | 25.4 c | 28.5 b | 32.3 a |
| Filamentous fungi | $10^4$ cfu g$^{-1}$ | 27.9 c | 40.6 b | 60.6 a |

The letters a, b, and c indicate significant difference at $p < 0.05$.

### 3.4. Experiment 4

Replacing ploughing, basic fertilisation, seedbed preparation, and row sowing (the CT system) with a single pass of a multifunctional machine (the ST-OP method) reduced water loss from the soil during the sowing of winter and spring crops. This increased the amount of water remaining in the soil. The soil water reserve in the 0–20 cm soil layer in the winter barley sowing period was 82 m$^3$·ha$^{-1}$ greater using the ST-OP method than for conventionally tillage (Figure 5). Replacing conventional tillage with the ST-OP method in the cultivation of maize increased soil water reserve during the sowing period by 97 m$^3$·ha$^{-1}$.

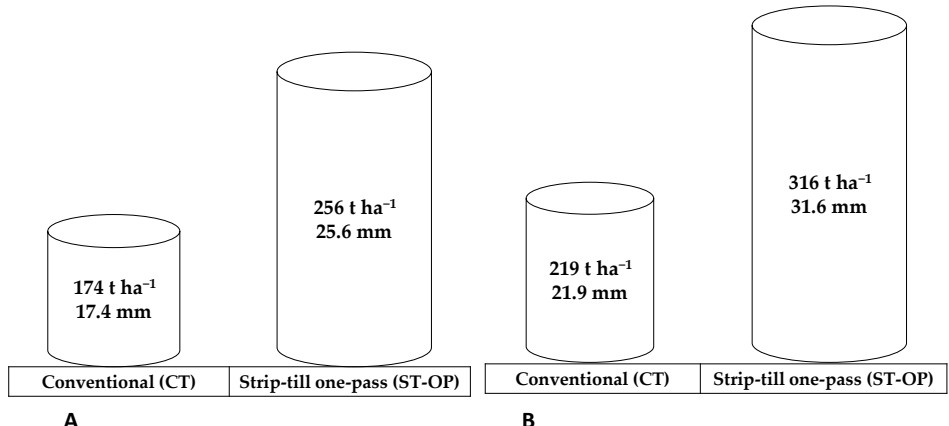

**Figure 5.** Soil water reserve in the 0–20 cm soil layer in the sowing period. (**A**) Winter barley (from 20 August to 20 September, 2016–2018); (**B**) Maize (from 20 April to 20 May, 2017–2019).

Immediately after tillage of soil strips and sowing winter barley in them using the ST-OP method, a spatial differentiation of soil moisture occurred (Figure 6). The greatest amount of moisture remained in the soil of the non-loosened inter-row, beneath the dry soil pushed from the seed strips to create the contour ridge. The absolute difference in moisture between the non-loosened layer and the soil of the contour ridge was 5.4% volume Moisture in the deeply loosened soil in the seed strips was significantly lower (2.1% by absolute volume) than in the soil in the inter-rows, but significantly higher (3.3% volume) than in the soil of the contour ridge insulating the inter-row. In relative terms, these differences in soil moisture were 16.8% and 46.5%, respectively. After sowing maize, soil moisture in the inter-row was higher than in the seed strips by about 2–3% volume in absolute terms (16–20% higher in relative terms) (Figure 7).

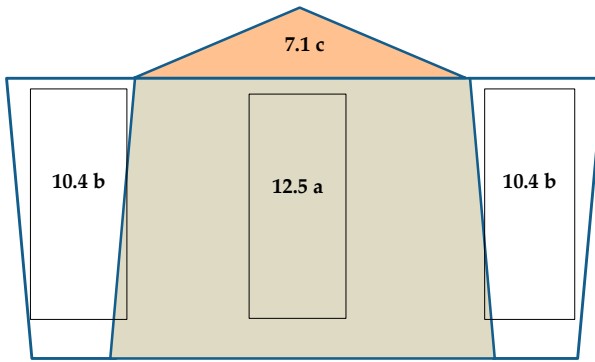

**Figure 6.** Spatial differentiation of soil moisture (% *v/v*) immediately after sowing winter barley. The letters a, b, and c indicate significant difference at $p < 0.05$.

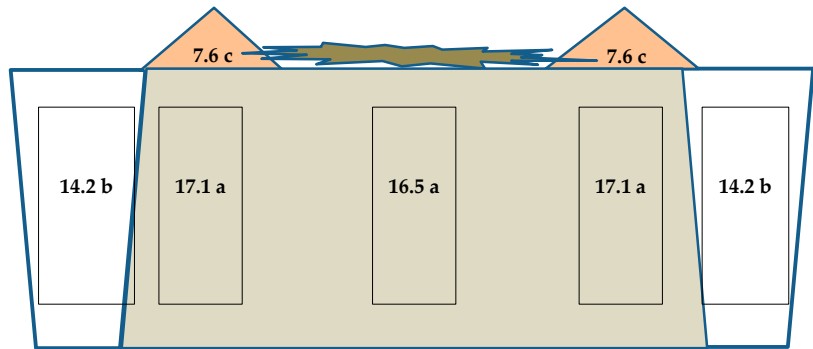

**Figure 7.** Spatial differentiation of soil moisture (% *v/v*) immediately after sowing maize. The letters a, b, and c indicate significant difference at $p < 0.05$.

As compared with CT, the ST-OP method reduced labour input, fuel consumption, and atmospheric $CO_2$ emissions (Table 5). Savings in the cultivation of 1 hectare of winter barley amounted to 114 min of labour, 35.9 litres of diesel, and 98.7 kg of $CO_2$. Savings in the cultivation of 1 ha of maize were even greater, i.e., 152 min, 45.8 litres of fuel, and 125.9 kg of $CO_2$.

**Table 5.** Inputs and $CO_2$ emissions as a result of different tillage systems. Conventional (CT), reduced (RT), and strip-till one-pass (ST-OP).

| Crop | Unit | Conventional (CT) | Strip-Till One-Pass (ST-OP) |
|---|---|---|---|
| | | labor input | |
| Winter barley | min ha$^{-1}$ | 142 a | 28 b |
| Maize | min ha$^{-1}$ | 175 a | 23 b |
| | | fuel consumption | |
| Winter barley | L ha$^{-1}$ | 49.7 a | 13.8 b |
| Maize | L ha$^{-1}$ | 58.0 a | 12.2 b |
| | | CO2 emissions | |
| Winter barley | kg ha$^{-1}$ | 136.7 a | 38.0 b |
| Maize | kg ha$^{-1}$ | 159.5 a | 33.6 b |

The letters a and b indicate significant difference at $p < 0.05$.

## 4. Discussion

At the beginning of the 21st century, it was estimated that the area of agricultural soils tilled following the principles of conservation agriculture might increase by nearly 5 million hectares in the near future. An important feature of this approach is reduced tillage, including mulch-tillage,

ridge-tillage, no-tillage, and strip-tillage [48]. Therefore, the present researchers' research on the impact of a ST-OP system on the enviroment, mainly soil, is of relevance to this topic. Another reason for undertaking the research was the relative scarcity of scientific findings regarding strip-till as a soil tillage system in conservation agriculture, especially in Europe [49,50].

The beneficial effect of reduced tillage methods on soil properties results from, among other things, the large amount of plant residues left on the field surface and functioning as a mulch. According to the ECOFYS report [51], after harvesting high-yield crops, 5.92–11.75 t·ha$^{-1}$ of plant residues remain. In the habitat and agricultural conditions in which the present research was carried out, the amount of plant residues left after harvesting the main crops was even greater, especially for maize and sunflower residues, i.e., there remained 15.65 t·ha$^{-1}$ and 10.23 t·ha$^{-1}$, respectively, on the soil surface. Plough tillage turns the soil over and effectively covers the debris. In our own research, about 3–5% of residues remained on the soil surface. More residues are left with reduced tillage, although the amount depends on the particular method used. According to Mairhofer et al. [52], after tilling with a cultivator, about 10–25% of residues remained on the surface, depending on depth of tillage. Tillage and sowing using the ST-OP system with 36 cm strip spacing, as carried out in our research, resulted in 62.7–74.7% of residues remaining on the field surface after wheat, and 75.7–82.0% after maize. Therefore, these amounts are closer to the amounts of residue left on the field surface after no-till and direct sowing than after conventional tillage. The benefits that large amounts of plant residue on the field surface have for soil quality and protection are scientifically documented. The literature highlights an effect on, among other things, soil water conditions and surface runoff, soil erosion, nutrient availability, content and transformations of organic carbon, weed infestation, and soil organisms [53].

The ST-OP system has proven to be a method that effectively reduces soil loss on a slope of 11% gradient. In the long growing seasons of winter rape and winter wheat, 470 mm and 560 mm, respectively, of precipitation were recorded. In the third year, despite maize being grown with wide row spacing, in its shorter growing season the sum of precipitation was only 240 mm. A direct proportionality between intensity of precipitation and amount of surface runoff and loss of soil on a slope was indicated by the studies of Zhao et al. [54]. They also showed, in a model experiment, that the amount of water erosion and soil loss depended on the vegetation on a slope. The soil tillage method is equally important in the agricultural use of soils exposed to erosion. It affects water infiltration, runoff intensity, and soil loss [55]. Reduced tillage without ploughing but leaving large amounts of plant residue on the surface, and especially no-till, all reduce erosion and soil loss. Chowaniak et al. [56] also conducting research, in a climate appropriate for Poland, and stated that the loss of soil under the influence of no-till cultivation was 66.8% lower than with conventional tillage. The high effectiveness of strip-till for reducing water erosion (similar to the results of our own research) is indicated by Laufer et al. [57]. On the basis of a comparison of strip-till, reduced tillage, and conventional tillage, they found that the soil loss after strip-till cultivation was 92% less than after conventional tillage, which was due to increased water infiltration.

Plant residues on the surface reduce surface runoff and soil loss [58,59], and also protect soil aggregates and soil structure. According to Zheng et al. [60], no tillage and spacing tillage increased the proportion of soil macro-aggregates as compared with mouldboard ploughing and conventional tillage. Spacing tillage increased the proportion of water-stable aggregates with a diameter >0.25 mm by 34.5% as compared with other tillage systems, which may be a good way to improve the durability of the soil structure. In our own research, the ST-OP method increased the share of large diameter aggregates (7–10 mm) by 57.7% as compared with conventional tillage. The relative increase in stability of aggregates of diameter 0.25–2.0 mm in soil thusly cultivated as compared with plough tillage was 24.2%, and 28.8% for aggregates with a diameter of 2–10 mm. Al-Kaisi et al. [61] compared the impacts on soil structure under five tillage systems (including zero tillage, strip-till and plough tillage) and found greater micro-aggregate and macro-aggregate stability under the influence of zero tillage and strip-till, although the difference as compared with intensive systems, including plough tillage, was not large.

Biological properties, including the presence and activity of microorganisms, are an important factor shaping the soil structure and its durability. According to Chotte [62], the participation of microorganisms in the formation and stability of soil aggregates is a very complex process. It involves bacteria, fungi, and plant roots, and their secretions. The presence and activity of earthworms is no less important [63]. These organisms participate in the formation of aggregates and a durable structure by influencing the physical and chemical properties of the soil. According to the research results of Hallam et al. [64], their presence in soil increased water-holding capacity, plant-available water, organic matter content, and water-stable aggregates of >0.250 mm. The relative differences in these features over soil with no earthworms present was 9–21%. In the present research, the many times greater number and mass of earthworms found under the influence of the ST-OP system as compared with RT, and especially as compared with CT, may, in addition to the greater abundance of bacteria and fungi, explain the greater soil aggregate stability. The greater stability of the structure may also result from the higher content of glomalin (EEGRSP). According to Rillig [65], soil aggregate stability (SAS) correlates positively with glomalin content. That author's research found that the correlation coefficient (r) between easily extractable glomalin-related soil protein (EEGRSP) and SAS was 0.58 to 0.84. In turn, Wright et al. [34] found that glomalin content in soil and SAS increased after switching from plough tillage to reduced tillage.

In our own research, reduced tillage combined with simultaneous basic fertilisation and sowing (the ST-OP method) increased soil water reserve in the sowing period for winter and spring crops. This is advantageous because the research site, and much of Poland, are located in areas of frequent rainfall deficits and dry periods [66,67]. Replacing conventional sowing of plants after separate ploughing, to instead use a one-pass technology, resulted in water savings during the soil preparation and sowing periods for winter and spring crops that corresponded to 8.2 mm and 9.7 mm of rainfall, respectively. These are significant amounts, corresponding to the sum of precipitation for about 1–2 weeks of the early spring, winter, or late autumn months in the study area. According to scientific studies, the beneficial effect of ST-OP technology on soil moisture and soil water reserve as compared with plough cultivation is the result of plant residues left as mulch on the surface reducing evaporation [68,69]. Plant residues reduce surface runoff and temperatures and increase water infiltration and retention in the soil [70,71]. The increased water content in the soil also results from the lack of deep loosening tillage that heavily aerates soil [72]. According to Alvarez and Steinbach [73], the water content is higher in ploughed soil than in unploughed soil, and the difference may cover the 1 to 3 days of evapotranspiration during the crops' intensive growth and flowering periods. The relationship of bulk density and porosity to soil moisture [74], and the effect of plant residues on soil properties [53], may explain the spatial variation in soil moisture after the ST-OP method, and especially the fact that moisture is greater in the inter-row zone covered with loose soil and/or mulch than in the loosened rows.

The conducted research confirmed the results of previous experimental studies on the potential for reduced inputs (including labour, fuel consumption, and atmospheric $CO_2$ emissions) by replacing conventional soil tillage and sowing technology with the strip-till method. The research concerned plants grown in rows with both wide and narrow spacings [75,76]. The savings were probably derived from deep soil loosening being limited to narrow strips covering only about one third of the field surface. The several treatments that conventional methods require for basic soil tillage, seedbed preparation, fertilization, and seed sowing were replaced with a single pass of a hybrid machine implementing the ST-OP method. For the above reasons, the labour input into maize cultivation was eight times less, and fuel input and $CO_2$ emissions were nearly five times less.

## 5. Conclusions

The results of experimental field studies, which included a long-term static experiment, show that a strip-till one-pass system can be treated as a component of conservation agriculture. The lack of mechanical treatments after harvesting the pre-crop, and the loosening of only narrow strips of soil, mean that, after sowing the subsequent crop, a large amount of post-harvest residues remains on the

surface to protect the soil. In the conducted research, over 60–70% of pre-crop residues were left on the field surface. Plant residues on the surface and deep tilling of soil strips to enable water infiltration reduced surface runoff, reducing soil loss on a slope of 11% gradient by a factor of over six (6.36 t·ha$^{-1}$ year$^{-1}$) than after plough tillage. The ST-OP method resulted in soil containing more microorganisms (bacteria and fungi) as compared with conventional tillage. Moreover, the number of earthworms in the soil was over three times higher, and their mass was almost five times greater. Earthworm activity and glomalin content are factors that increase the durability of the soil structure. The long-term application of the ST-OP method resulted in glomalin contents (EEGRSP) in the soil having increased by 9.4% and 4.5% relative to CT and RT, respectively. In the tested soil, the stability index (SAS) of aggregates of diameter 0.25–2.0 mm was significantly higher than after plough tillage, and aggregates with a diameter of 2.0–10.0 mm were more stable than aggregates of the same size in soil tilled either by ploughing or by reduced tillage. The favourable changes in soil properties resulted from the soil water reserve in the sowing period being 8.2 mm and 9.7 mm greater than under plough tillage for winter and spring plants, respectively. Introducing a strip-till one-pass method to larger-scale field crop production may measurably reduce labour and fuel inputs, as well as atmospheric $CO_2$ emissions. For a single hectare of maize, as compared with conventional ploughing, fertilization, and sowing, 45.8 l litres of fuel can be saved and $CO_2$ emissions can be reduced by 125.9 kg. Despite the favourable research results, further in-depth studies on changes in soil properties are needed, including on the balance of organic matter and a detailed economic and energy accounting of the use of this method of soil and plant cultivation.

**Author Contributions:** Conceptualization, I.J. and D.J.; methodology, I.J., D.J., and K.R.; investigation and performed the field experiments, D.J., I.J., and P.W.; data curation, compiled and analysed the results, I.J., D.J., and P.W.; writing—original draft preparation, I.J., D.J., and P.W.; review and editing, I.J., D.J., P.W., and K.R. All authors have read and agreed to the published version of the manuscript.

**Funding:** This research received no external funding.

**Acknowledgments:** The authors thank the company Agro-Land Marek Różniak at Śmielin, Poland, for allowing experimentation in their production field with the use of the strip-till hybrid machine Mzuri Pro-Til.

**Conflicts of Interest:** The authors declare no conflict of interest.

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
