# Peer review of "A Strip-Till One-Pass System as a Component of Conservation Agriculture"

_agronomy, doi:10.3390/agronomy10122015_

Round 1
Reviewer 1 Report
- 117 line. Would be more informative if meteorological conditions be presented in article but not reference "[43]" to other article.
- 130-131 lines. "(CT)" and "(RT)" do not exactly meet description which provide in chapter Abstract.
- Is not of aim many times repeat description and their meanings, example: "conventional tillage (CT)", "reduced tillage (RT)", "strip tillage one pass (ST-OP)" and etc.
- Need to supplement chapter of Materials and Methods. What was model of tractor used in research? What was tractor speed?
- I suggest rewrite sentence after tables 1, 2, and 3, by the way, after Fig. 6 and Fig. 7. My offer is: Different letters in rows indicate significant difference at p<0.05.
- Need write what was minimum significant difference limit In table 1, table 2, and Fig. 4.
- Table 4. Why is not provide results about other studied plants (winter wheat, winter rape and etc.).
Author Response
Thank you for your comments on the manuscript and suggestions for improvement. Corrections were written in red against the background of the first version of the manuscript - black font and comments as suggested by the second reviewer - blue font. Corrections made and response to review: - The weather conditions during the implementation of single 3-year field experiments appropriate for this study are presented in an additional table. The meteorological conditions during the multi-year static experiment and the average multi-year data were described in the previous paper [44]. - In accordance with the reviewer's note, a detailed correction was made in the Abstract. - The machine cooperated with the tractor Claas Axion 850, speed about 7-8 km/h. - The significance of the differences between the results presented in the tables and figures was marked with different letters as groups of homogeneous results. LSD was not given so as not to duplicate the significance markings. As suggested by the reviewer, the description of the group of homogeneous results under the tables and figures has been corrected. - Table 4 presents the results of experiment no. 4. In experiment no. 4, only two species of crops, winter barley and maize, were affected by different methods of soil tillage, fertilization and sowing.
Reviewer 2 Report
The manuscript reports the results of multiyear field experiments carried out in North-West Poland to study the effect of the strip till technique on some physical and biological soil properties and energy consumption. The paper itself reports results of interested, even if not completely new. However, the main constrain of the manuscript is that, according to my opinion, is not well structured and it is very difficult to read and understand.
Overall and because of this general constrain, I would suggest the Authors to deeply revise and resubmit it.
I have provided below some advices to improve the logical framework of the sections of the manuscript. I do hope these comments may be of guidance to the Authors. Once the major weaknesses will be solved, I would be pleased to reconsider the manuscript.
Introduction:
- I found it too long and without a clear, logical flow; I would suggest to shorten the part about the conservation agriculture in general and focus on the strip till, presenting pros and cons;
- also, the part related to the glomalin is too long and should be shortened. If opportune, some sentences, when appropriate, can be moved to the discussion section;
- in my opinion, the aims of the study are not well presented and are not really clear. Why the Authors have done this study? What were the critical/unknown aspects of strip till?
M&M
- the M&M section is very confused as well and should be restructured; be aware to seprate information related to the experimental design and those about the measurements: i.e. the sections “experiment 3” and “experiments 4” describe measurements done in sample collected in experiment 1 and 2 (then 3 and 4 are not experiments ?)
- in the section experiment 1 is not clear in which year the 7 crops have benne grown: what is the sequence? Then what is the time and the space arrangement of them?
- I would suggest dedicating an independent section or subparagraph to describe the strip till technique;
- According to ehat above reported, my suggestion is to reorganise the M&M section as follow: i) study site; ii) field experiment design; iii) strip till technique description; iv) measurements; v) statistical elaboration.
- the quality of figure 1 is not adequate and in its present form it is not really informative and could be omitted;
Results
- The results should be separately reported by experiments (exp. 1 and exp. 2). The synthesis obtained framing the outcomes of the two experiments should be instead reported in the discussion section;
- Many sentences (i.e. lines 269 – 270; lines 284 – 285; lines 336 – 338) of the result session should be moved to the discussion;
- I see that in the results session
Conclusions
- I noticed some redundancies with the discussion section
Author Response
Thank you for your comments on the manuscript and suggestions for improvement. Corrections were written in blue against the background of the first version of the manuscript – black font (Redundant sentences have been deleted) and comments as suggested by the first reviewer – red font. Corrections made and response to review: In line with the reviewer’s suggestion, the article was re-structured. Numerous changes were introduced that should make it more logical and comprehensible. Introduction The section was deeply restructured - text reduction, changes, additions, information sequence (the entire chapter is marked in blue). The information directly addressing conservation agriculture was reduced, as was the information relating to glomalin. The information on the strip-till method and its environmental and production-level impacts was put in order. The study’s premises, hypotheses and objectives were articulated more clearly. Materials and methods Clarification: The experiments designation as 3 and 4 in the manuscript were performed independently of experiments 1 and 2 as part of a study in a location of their own and with a separate duration. In the revised article the authors have tried to present this unambiguously. The methods applied in the four independent experiments that comprise the study have been presented in sequence. Thus, the previous form has been maintained but changes and additions have been introduced to make it clearer and more comprehensible. As suggested by the reviewer, Figure 1 was removed. The additional figure shows the scheme of experiment no 1 (example 1 with 3 replications). Results In accordance with the reviewer’s suggestions, the results of individual experiments have been reported in separate paragraphs labelled Experiment 1, etc. Some sentences have been moved from the results section to the discussion, and those not relating directly to the results have been removed.